# Artificial Intelligence, Wearables and Remote Monitoring for Heart Failure: Current and Future Applications

**DOI:** 10.3390/diagnostics12122964

**Published:** 2022-11-26

**Authors:** Nitesh Gautam, Sai Nikhila Ghanta, Joshua Mueller, Munthir Mansour, Zhongning Chen, Clara Puente, Yu Mi Ha, Tushar Tarun, Gaurav Dhar, Kalai Sivakumar, Yiye Zhang, Ahmed Abu Halimeh, Ukash Nakarmi, Sadeer Al-Kindi, Deeptankar DeMazumder, Subhi J. Al’Aref

**Affiliations:** 1Department of Internal Medicine, University of Arkansas for Medical Sciences, Little Rock, AR 72205, USA; 2Department of Internal Medicine, University of Arkansas for Medical Sciences Northwest Regional Campus, Fayetteville, AR 72703, USA; 3Department of Hematology and Oncology, University of Arkansas for Medical Sciences, Little Rock, AR 72205, USA; 4Division of Cardiology, Department of Internal Medicine, University of Arkansas for Medical Sciences, Little Rock, AR 72205, USA; 5Department of Population Health Sciences, Weill Cornell Medicine, New York, NY 10065, USA; 6Information Science Department, University of Arkansas at Little Rock, Little Rock, AR 72204, USA; 7Department of Computer Science and Computer Engineering, University of Arkansas, Fayetteville, AR 72701, USA; 8University Hospitals Harrington Heart & Vascular Institute, Case Western Reserve University School of Medicine, Cleveland, OH 44106, USA; 9Division of Cardiology, Department of Internal Medicine, Richard L. Roudebush Veterans’ Administration Medical Center Indiana Institute for Medical Research, Indiana University School of Medicine, Indianapolis, IN 46202, USA

**Keywords:** machine learning, heart failure, remote monitoring, pressure sensors, time-series analysis

## Abstract

Substantial milestones have been attained in the field of heart failure (HF) diagnostics and therapeutics in the past several years that have translated into decreased mortality but a paradoxical increase in HF-related hospitalizations. With increasing data digitalization and access, remote monitoring via wearables and implantables have the potential to transform ambulatory care workflow, with a particular focus on reducing HF hospitalizations. Additionally, artificial intelligence and machine learning (AI/ML) have been increasingly employed at multiple stages of healthcare due to their power in assimilating and integrating multidimensional multimodal data and the creation of accurate prediction models. With the ever-increasing troves of data, the implementation of AI/ML algorithms could help improve workflow and outcomes of HF patients, especially time series data collected via remote monitoring. In this review, we sought to describe the basics of AI/ML algorithms with a focus on time series forecasting and the current state of AI/ML within the context of wearable technology in HF, followed by a discussion of the present limitations, including data integration, privacy, and challenges specific to AI/ML application within healthcare.

## 1. Introduction

Despite progress in understanding the pathophysiology of heart failure (HF), it remains a significant contributor to global morbidity and mortality. While continuous efforts to improve the understanding of HF over the past two decades have translated into a reduction in mortality, HF-related hospitalizations have increased over time [1]. Moreover, with increasing complexity of HF patients and an increasing proportion of these patients receiving implantable devices, the consequent costs incurred by the United States healthcare system was $43 billion in 2020 alone and are projected to increase to $78 billion by 2030 [2,3]. Inpatient hospitalizations, contributing around 80 percent of the total medical costs for HF, have critical prognostic implications [4]. HF hospitalizations are also associated with a high readmission rate, with nearly 1 in 5 patients getting readmitted within 30 days and every other patient getting rehospitalized by 6 months [5,6]. Therefore, there is a pressing need for novel approaches to reduce the incidence of HF-related hospitalizations [2,7]. While guideline-directed medical therapy coupled with close follow-up can reduce the incidence of HF hospitalizations, the paucity of widely available medical resources and workforce limits the ability to tackle the issue of HF hospitalizations and readmissions [8].

The use of wearable device technology and implantable cardiac devices have ushered in a new approach in the field of cardiovascular diagnostics in recent years, offering healthcare providers with an opportunity to monitor disease dynamics remotely and creating a window of opportunity to intervene in a timely and more effective manner—thereby potentially avoiding hospitalizations secondary to hemodynamic decompensations and adverse cardiac events [9,10]. This can translate into increased healthcare system productivity in two ways (i) a higher proportion of individuals can be intervened upon remotely, leading to better outcomes, and (ii) medical resources can be judiciously diverted to individuals with advanced HF who are prone to frequent decompensations, thereby improving overall outcomes.

While the current state of wearables aims to reduce adverse cardiac events, it can be potentially utilized in a multitude of ways owing to the unique nature of the remotely collected data. Clinical variables are transmitted longitudinally over time, and the massive chunks of timestamped data can be analyzed to study disease trajectories and identify impending cardiac events. Moreover, when assembled from multiple patients, these variables can be used to phenogroup similar individuals, identify high-risk patients, develop risk and survival models, and discover new variables imparting worse prognoses. The current statistical models are limited by their ability to include finite datasets and the scientific assumptions inherent to the statistical model used. Conversely, artificial intelligence and machine learning (AI/ML) approaches require minimal assumptions, can illustrate complex, non-linear relationships as often observed in biologically derived data, and incorporate multiple variables for data modeling.

Given the above advantages, AI/ML models have been tested widely and have shown promising results when studied in conjunction with their statistical counterparts in the field of cardiovascular medicine [11]. With the evolution of AI/ML and the dawn of digital health in HF, AI/ML can be envisioned to play a significant role in improving the current understanding and monitoring of HF. In this review, we sought to introduce the concepts of AI/ML algorithms and time series analysis, the current state of wearable devices in HF, and the potential applications of AI/ML in wearable technology. Finally, we describe the limitations at present in adopting AI/ML technology, the challenges with digital data sharing and integration in healthcare, and discuss potential solutions.

## 2. Introduction to AI/ML and Time-Series Forecasting

As described by Arthur Samuel, ML is a field of study that provides learning capability to computers without being explicitly programmed [12]. Learning happens when computers are fed with large amounts of data and a known output (i.e., training data). It often requires high computational power but can unravel non-linear, implicit relationships between the fed variables and the outcome of interest. Once trained, these computational algorithms can process unseen data to predict meaningful outcomes. ML has been applied widely in cardiovascular medicine in the last few years and can broadly be classified into unsupervised and supervised ML (Figure 1).

Unsupervised ML is a branch of ML whereby structured data, without any data labels, is fed into an algorithm to understand hidden relationships within data points. Data mining can be done to cluster similar data points into various groups, which can be used for phenogrouping (clustering algorithms such as k-means, hierarchical, etc.). Alternatively, association algorithms can be employed to help find relationships between data points themselves (Apriori and Equivalence Class Clustering and bottom-up Lattice Traversal (ECLAT) algorithms). Association algorithms have been applied vastly in market basket analysis to analyze customer purchasing patterns and have been recently employed in medicine. Moreover, unsupervised ML can also condense multiple data points into fewer dimensions (dimensionality reduction) via approaches such as principal component analysis. Supervised ML, on the other hand, requires an initial data labeling of the structured data, which is then fed through training algorithms to compute relationships between variables and an outcome of choice. The algorithms can be trained to either classify an output to predefined discrete categories (classification algorithms) or quantify relationships between variables (regression algorithms) [14]. The above-mentioned techniques have been applied in the field of HF for initial diagnosis, modeling disease prognoses, goal-directed medical therapy (GDMT) optimization, predicting outcomes for device therapy interventions, etc. (Figure 2) [13].

Deep learning (DL) is a branch of AI/ML that can process large complex data, such as those derived from medical images or videos [15]. DL usually takes the form of neural networks consisting of multiple nodes (i.e., neurons) arranged into an input, multiple hidden layers, as well as an output layer. When a trained neural network is fed with data, specific neurons in the input layer are activated, which pass the command to a chain of neurons in the hidden layers and then the output layer; the command becomes increasingly complex as the information is passed along successive layers. Training of DL algorithms can be done in both unsupervised and supervised fashion and has been used widely in cardiovascular medicine to automate medical images interpretation, create multimodal prognostic models, as well as uncover unknown and undiscovered associations between phenotypes and a particular condition/outcome [15]. For example, a DL algorithm has also been shown to be superior to sonographers for assessing left ventricular ejection fraction on 2-D echocardiography in the recently concluded EchoNet randomized controlled trial [16].

While these AI/ML techniques have been applied in the field of cardiovascular medicine, traditionally, data analysis via AI/ML usually has not taken into account the dimension of time. Continuously transmitted data via remote monitoring at regular intervals gives the opportunity to plot variables as a function of time and forecast future variables based on the prior trends, referred to as time series forecasting [17]. The concept of time series forecasting is not new and has been applied in fields outside of medicine. Modeling can be done to forecast the value of one variable independently (univariate) or as a function of multiple dependent variables (multivariate) with time. Statistical models developed for time-series analysis posit multiple assumptions about the data. Stationarity is a vital concept in time-series analysis, which implies that the data’s statistical properties (e.g., mean, variance, etc.) do not change with time. Real-world data is usually non-stationary, exhibiting trends, seasonality, or cyclical patterns when plotted chronologically [17].

While traditional statistical models such as Autoregressive moving average (ARMA) necessitate the time-series data to be converted to stationary data before model construction via approaches such as differencing and detrending, newer statistical and AI/ML models can be employed with non-stationary time series [17,18]. Moreover, recurrent neural network (RNN)-based approaches that can handle large amounts of non-linear data have been described, which could prove ideal for the terabytes of data transmitted via remote monitoring. In contrast to traditional neural networks, which are strictly feed-forward neural networks, the output of the hidden layers of neurons can be fed back into neurons in RNNs, which is used in time-series forecasting [19]. This hidden state output is then successfully combined with new input to generate the final output. During the training phase, the predicted output is then compared with the ground truth to calculate the error. Backpropagation is then done to calculate the gradients and adjust the weights of the layers for accurate prediction. RNNs might suffer from the ‘vanishing’ or ‘exploding’ gradient problem due to an exponential decrease or increase during backpropagation, respectively, leading to decreased performance of the model. Hence, RNNs can function well with a limited set of sequential data, with decreasing accuracy with increasing data [19]. To solve this problem, Long short-term memory (LSTM) and gated recurrent units (GRU) models have been described, which can retain longer sequential data by the virtue of ‘gates’ [20]. Details about AI/ML algorithms used for time series forecasting are provided in Table 1.

## 3. The Current State of Telemonitoring, Wearable, and Implantable Device Technology in HF

Trials aimed at monitoring patients via telephonic interventions and wearables have been attempted in the last decade, albeit with limited success [21,22,23]. The Telemedical Interventional Management in Heart Failure (TIM-HF) trial enrolled 710 patients with a left ventricular ejection fraction (LVEF) of ≤35%, NYHA class II-III symptoms, and a 2-year history of HF decompensation and found no differences between telemonitoring (weight, blood pressure, and heart rhythm data transmission) and standard care in cardiovascular and all-cause mortality or hospitalizations at a median follow-up of 26 months [22]. More recently, the TIM-HF2 trial showed superior results for web-based remote patient management than routine care in reducing HF hospitalizations and associated mortality [23]. In contrast to TIM-HF, the TIM-HF2 study excluded patients with depression and recruited patients who had been hospitalized in the last 12 months. Of note, the investigators also broadened their selection criteria to include patients with LVEF ≤ 45% or LVEF ≥ 45% on diuretics. Their structured remote intervention was associated with a smaller proportion of days lost from unplanned HF-related hospital admissions (HR 0.80, 95% CI 0.65–1.00; *p* = 0.0460) and had lower all-cause mortality (HR 0.70 [95% CI 0.50–0.96], *p* = 0.28) compared to usual care [23]. Furthermore, the Better Effectiveness After Transition- Heart Failure (BEAT-HF) trial, utilizing predischarge HF education coupled with regular telephone visits and telemonitoring, did not show any statistical difference in reducing 180-day readmission rates in patients with HF (HR: 1.03 [95% CI, 0.88–1.20]; *p* = 0.74) [21]. Although telemonitoring was the backbone of the above trials, the recent emergence of wearables and novel sensor devices that can accurately detect intrinsic hemodynamic changes have demonstrated the potential to change the remote monitoring landscape.

Most representative studies of wearables in HF patients have involved pedometers and actigraphy devices which, by analyzing the pattern of physical activity, can enable patients and physicians to engage in lifestyle modification and therapy modulation [24,25]. Moreover, several emerging technologies have been tested in monitoring HF readmissions. While thoracic impedance monitoring via implantable cardioverter defibrillators (ICDs) has been shown to be a better predictor than daily weight monitoring for assessing pulmonary fluid status, studies of impedance-derived fluid index (Optivol index) for predicting HF hospitalizations in the ambulatory setting have yielded conflicting results [26,27,28,29,30]. Remote dialectic sensing (RedS) has been recently proposed for monitoring pulmonary congestion in HF. In a recent analysis, patients were provided with a vest, which they wore for a period of 90 s every day. Electromagnetic analysis of tissue dielectric properties, reflecting the degree of pulmonary congestion, was used to guide clinical management. ReDS-guided management decreased 90-day readmissions by 87% (HR: 0.07 [95% CI: 0.01–0.54], *p* < 0.01). Notably, discontinuation of ReDS increased readmissions by 79% (HR: 0.11 [95% CI: 0.012–0.88], *p* < 0.05) [31].

Similar to wearables, implantable devices, such as pacemakers, ICDs, and intracardiac pressure sensors, also generate an enormous amount of data that can be used to create algorithm-based risk prediction models aimed at early diagnosis treatment selection. For example, the MultiSENSE (Multisensor Chronic Evaluation in Ambulatory Heart Failure Patients) study tested the HeartLogic multisensor index and alert algorithm for predicting HF exacerbation [24]. In 900 patients receiving cardiac resynchronization therapy, the HeartLogix algorithmic analysis of sensor data (the first and third heart sounds, thoracic impedance, respiration rate, ratio of respiration rate to tidal volume, heart rate, and patient activity) yielded a sensitivity of 70% (95% CI: 55.4–82.1%) for early detection of HF exacerbation with an unexplained alert rate of 1.47 (95% CI: 1.32 to 1.65) per patient-year, the median time from alert onset to a HF exacerbation being 34 days (interquartile range: 19.0 to 66.3 days). In addition, the algorithm’s performance was assessed across a range of thresholds, providing an opportunity to customize its use for the individual patient.

The latest addition to the family of remote monitoring devices has been the implantable pulmonary artery and the left atrial pressure sensors, with the benefits of remote monitoring of pulmonary artery pressures (PAPs) in patients with HF being confirmed in multiple trials [9,10,32]. More recently, with the VECTOR-HF trial showing positive results on the safety and usability of a digital, leadless intra-atrial pressure sensor (V-LAP system) in HF patients, the field is bound to grow [33]. Moreover, with data being transmitted via multiple digital devices, parameters can be clubbed together via AI/ML techniques to tailor to better outcomes. Although remote PAP, LAP, and other monitoring devices have high costs and technical limitations, this can also be amenable to an AI/ML solution by developing algorithms from previous PAP-monitored patient data to ‘learn’ who benefits most, thus enabling targeting the intervention to a specific population.

## 4. Current Applications of AI/ML in Remote Monitoring via Wearables and Implantable Cardiac Devices

While AI/ML has increasingly been used for detecting and diagnosing a variety of cardiac diseases, its application to remote monitoring data is relatively new. While remote monitoring has been utilized for the detection of cardiac arrhythmias and for coronary vascular events, cutting-edge clinical applications in the realm of HF include the use of artificial neural networks trained with cardiac function parameters (QT interval, heart rate variability, seismocardiogram signals) from wearables to accurately identify early decompensation of HF, identify patients at higher risk of sudden cardiac death (SCD), etc. (Table 2) [34,35,36,37,38,39,40,41,42,43]. Accurate identification of pathophysiological dynamics may be helpful as it allows for the institution and optimization of medical therapies before the onset and progression of symptoms, thereby reducing the risk of HF hospitalizations and readmissions. Additionally, early identification of patients at high risk of SCD may allow for earlier intervention, translating into reduced morbidity and mortality.

Novel wearable devices have been integrated with AI/ML, with the goal of investigating surrogates for pulmonary vascular congestion, thereby leading to earlier detection of clinical decompensation. A pilot study on 45 patients by Inan et al. investigated the feasibility of wearable seismocardiogram patches to proactively assess and manage patients with HF [35]. The researchers used wearable seismocardiogram patches to measure vibrations of the chest wall in response to each heartbeat. At-home hemodynamic changes were estimated with the 6-min walk test. The authors used an analytic graph mining–based approach to then quantify and compare the similarity of the structure of seismocardiogram signals before and after the 6-min walk test (graph similarity score). Since the decompensated (hospitalized) HF patients were unable to meet the increased cardiac requirement from the walking test, the changes in the graph similarity score (GSS) for decompensated HF patients were significantly lower than for compensated (outpatient) HF patients. This study showed that a small, wearable device could be used to identify HF states and preemptively treat patients before they required hospitalizations. More recently, using seismocardiogram-derived signals in 20 patients, good accuracies were reported for vasodilator-induced acute changes in mean pulmonary artery pressure (mPAP) and pulmonary capillary wedge pressure (PCWP) by the same group (root mean squared error of 2.5 mmHg and 1.9 mmHg for changes in mPAP and PCWP in the training dataset and 2.7 mmHg and 2.9 mmHg in the validation data sets, respectively), thus paving the way for easier tracking of monitoring of intracardiac hemodynamics [36]. Moreover, a recent hierarchical clustering algorithm developed by Burton et al. aiming to identify phenogroups with elevated left ventricular end-diastolic volume (LVEDV) used electromechanical data collected via photoplethysmography and CorVista phase space analysis, thereby providing insights into the characteristics leading towards phenotypes of myocardial dysfunction [47].

Furthermore, Stehlik et al. investigated the accuracy and applications of noninvasive wearable monitoring in detecting HF exacerbations [39]. A total of 100 subjects (74 with HF with reduced ejection fraction and 26 with HF with preserved ejection fraction) were fitted with a wearable sensor that collected continuous ECG waveform, skin impedance, temperature, and patient activity data. Cloud-based analytic platforms used similarity-based modeling to evaluate the cumulated study subject data. The personalized AI/ML platform and wearable sensor predicted hospitalization secondary to HF exacerbation within a 10-day positive window with a sensitivity of 76% and a specificity of 85%, thus demonstrating the viability of timely detection and early intervention of HF with the use of AI/ML and wearables.

Apart from predicting HF decompensation, research has also focused on predicting sudden cardiac death (SCD) events in patients with HF [38,44,45,46]. Acharya et al. developed a sudden cardiac death index (SCDI) utilizing discrete wavelet transform (DWT) and nonlinear AI/ML analysis of ECG signals four minutes before SCD onset [38]. The ECG signals were retrospectively obtained from Holter monitor data and the normal sinus rhythm database. At one-minute intervals, nonlinear features were isolated from DWT decomposed ECG signals. Support Vector Machines, Decision Trees, and k-Nearest Neighbors were used to differentiate between normal variant ECG signals and ECG signals associated with SCD subjects. The most distinguishing features were then combined to form a single integrated index (i.e., SCDI), which was able to predict impending SCD with an accuracy of 92.1%, 98.7%, 93.4%, and 92.1% at one, two, three, and four minutes, respectively. More recently, using a prospective multicenter registry, Kim et al. used pacemaker-collected data to predict clinically relevant atrial high-rate episodes (AHREs). With data from 721 patients without a history of atrial fibrillation or flutter, AI/ML-based models were able to outperform logistic regression-based models in predicting clinically relevant AHREs (AUC for logistic regression and XgBoost 0.669 and 0.745, respectively) [46].

Additionally, DeMazumder et al. used ML algorithms (EntropyX) at the signal processing level to adapt to the signal noise and extract physiological dynamics with very high accuracy and precision. In a prospective multicenter observational study of 852 HF patients in sinus rhythm, the EntropyX of cardiac repolarization (EntropyX_QT_) strongly predicted appropriate ICD shock as well as all-cause mortality, above and beyond a comprehensive set of conventional predictors and risk scores [48]. Unlike conventional measures of variability, EntropyX analysis of a time series does not require equally sampled time intervals or preprocessing of data and is less sensitive to the presence of outlying points such as ectopic beats or noise. EntropyX analysis of physiological dynamics provides unique insight into subclinical physiological deterioration such as early changes in metabolic and redox properties that cannot be detected by contemporary clinical measures [49]. This new paradigm has also proven important for highly accurate rhythm discrimination, including a ROC area under the curve of 98% for identifying very brief paroxysms of atrial fibrillation and outperforming ICD rhythm discrimination algorithms in a prospective multicenter study [50].

More recently, the DeepEntropy algorithm was developed for ML analysis of sinus rhythm during sleep to identify ostensibly healthy adults who will and will not develop new-onset atrial fibrillation over the next ten years. In a multicenter derivation study of 2807 community adults, the incorporation of DeepEntropy into a multivariate prediction model added major independent prognostic value to conventional risk factors and scores for predicting the incidence of new-onset AF [51]. Whereas the performance of conventional risk predictors are limited in diverse populations, a subsequent validation study in the Multi-Ethnic Study of Atherosclerosis demonstrated that DeepEntropy performs well regardless of age, sex, race/ethnicity, etc., and has potential for broad application [52]. Ultimately, randomized controlled trials are needed to definitively establish the clinical utility of AI/ML analysis of physiological time series for improving outcomes. With the initial success of wearable technology and the incorporation of AI/ML in pilot studies, multiple trials are underway employing AI/ML to further widen the scope of wearables in HF management, such as GDMT optimization, medication adherence, etc. [53,54,55,56,57,58,59,60] (Table 3). Moreover, innovative AI/ML algorithms are currently being designed utilizing ECG and acoustic recordings recorded by electronic stethoscopes, which can accurately detect changes in pulmonary artery pressures in patients with an implanted CardioMEMS device [55]. If successful, such an approach allowing for accurate monitoring of HF dynamics via wearables can help alleviate the burden on outpatient healthcare providers while enabling the creation of ‘personal’ patient profiles. Analysis of troves of remotely monitored individual patient data can allow for delineation of daily patterns in activity, medication adherence, cardiac hemodynamics, etc., which will help patients by allowing for the creation of personalized health management plans, and simultaneously physicians by providing zettabytes of reliably collected real-world data, which can be used to further HF research.

## 5. Limitations and Future Prospects

While the applications of AI/ML and digital health in the field of HF are enticing, noteworthy limitations exist that need to be addressed. Limitations exist at the level of data extraction, storage, privacy, and integration, with an added layer of challenges specific to the AI/ML model development, accuracy, and implementation.

### 5.1. Data Extraction and Storage

Currently, medical information is stored in local data centers with a fixed computational capacity which requires extensive resources for storage expansion, data transmission, and integration across platforms. With remote monitoring likely to play a significant role in guiding cardiovascular care in the future, cloud computing solutions can bridge the unmet need for data storage and integration solutions for such large amounts of data. Cloud computing can accommodate terabytes of data while allowing for quick data retrieval from multiple sources (remote monitoring data, EHRs, medical images, etc.) (Figure 3) [61]. Apart from integration at the patient level, ‘data decentralization’ can help medical researchers by potentially creating more extensive repositories of data for improved disease modeling and forecasting. Moreover, while tons of data are stored in electronic health records, such troves of data are not readily employable and often need to be converted and restricted before being fed into an ML algorithm. More recently, Aslan et al. devised a convolutional neural network (CNN) based approach to convert numerical data into images, which can then be accurately classified for the purpose of diagnosing HF [62]. Such an approach, along with data extraction methods that use natural language processing algorithms, can help expand the current data pool and thereby lead to improved performance of the current diagnostic and prognostic models [63].

### 5.2. Data Quality Challenges

Data privacy, sometimes also referred to as information privacy, is an area of data protection that concerns the proper handling of sensitive data, including personal data or other confidential data. Researchers are confronted with the issue of privacy when transmitting data over the internet. Data transmission over the internet is susceptible to cyber-attacks, misuse, tampering, etc., and technologies ensuring data encryption and safe data transmission need to be established. Blockchain technology, though initially introduced in cryptocurrency, can facilitate data decentralization, interoperability, and data integrity across institutions [64]. Electronic computer code-based agreements (i.e., smart contracts) can help improve data transparency and allow for easier data sharing between parties while ensuring data integrity, owing to the blockchain-based timestamped nature of the data [65].

There is also a lack of regulations on using patient health information generated via consumer-owned wearables by third-party companies. Confidentiality is paramount regarding data transmission and sharing, with data sharing being protected by the Health Insurance Portability and Accountability Act (HIPAA) regulations [66]. The HIPAA regulations currently cover physicians and associated business entities, but concrete guidelines are necessary to direct data handling by third-party companies [67].

### 5.3. Challenges to Digital Technology Adoption

There is a potential for a ‘digital divide’ with the dawn of wearable sensors in HF. Apart from the clinical heterogeneity observed in patients with HF, socioeconomic status (SES) plays a paramount role, with studies showing more than a 50% increase in HF risk with lower SES [68,69]. While wearable technology has the potential to transform cardiovascular care, it can increase health inequity, owing to the need for adequate digital affordability and literacy associated with wearable technology. For instance, in a survey of 4551 adults in the United States, individuals with a higher annual household income (defined as $75,000 or more per year) were more likely to use wearables than individuals with a household income of less than $20,000 per year (OR 2.6; 95% CI [1.39–4.86], *p* < 0.001) [70]. Incorporating device-related costs into healthcare insurance benefit packages is crucial for narrowing the digital divide based on economic disparities. Moreover, loaner digital wearables can help mitigate the financial implications on healthcare, provided checks are in place to ensure the return of the digital devices. The novel iShare program utilizing loaner iPhones and smartwatches in 200 patients demonstrated augmented participation of patients without access to smartphones in remote monitoring following an acute myocardial infarction. However, with a modest return percentage of 72% at the end of the 30-day follow-up period, a system of checks and incentives is needed for its widespread implementation [71]. Nevertheless, the cost-effectiveness of wearable devices coupled with AI modalities needs to be studied in HF, with such devices being previously proven to be cost-effective in the screening of other cardiac pathologies, such as atrial fibrillation [72]. With pulmonary arterial pressure monitoring being shown to be cost-effective in multiple studies, complementing remote monitoring with AI is likely to improve the cost–benefit ratio, although it yet needs to be proven across a spectrum of population cohorts [73,74].

A lack of digital literacy can not only impede the adoption of digital technology but can also impact patient compliance with data transmission. Not surprisingly, individuals possessing college degrees or greater have been shown to be more receptive to the adoption of digital technology when compared to people with a lesser educational background [70]. Moreover, elderly individuals are less likely to adopt the usage of digital technology and also to comply with regular data transmission [70]. While it is essential to devise ways to improve the usage of wearables in this at-risk population, how these measures can be implemented in patients with neurocognitive disorders and visual/hearing impairment remains challenging.

### 5.4. Challenges Inherent to AI/ML Model Development and Processing

Apart from intensive computational power, model training requires diversely represented data and well-thought-of model assumptions. If overlooked, the above factors risk introducing implicit selection bias in terms of gender, race, socioeconomic status, etc. [75]. For instance, Obermeyer et al. described their observations on a healthcare algorithm aimed at recognizing the medical needs to be preferentially skewed towards white patients. By using healthcare spending as a proxy for the severity of medical illness, the algorithm underestimated the number of sicker black patients by more than half [76]. Careful data curation and representation in the model pre-processing stage can help mitigate some of the inherent bias.

While AI/ML models have the advantage of incorporating more horizontal data, the model’s predictive power starts decreasing with increasing dimensional complexity, often referred to as the ‘Hughes phenomenon’ [77]. Multi-dimensional data, when fed, leads to a lower degree of bias but a high variability. This leads to higher accuracy when tested internally but a dismal external performance. Smaller retrospective databases are especially susceptible to overfitting, given the higher rate of data missingness and irregularities, leading to additional noise being incorporated into the model. Moreover, data mining approaches on national database registries face the unique challenge of a high degree of dimensional heterogeneity, predisposing to overfitting. While solutions such as k-fold cross-validation and regularization have been proposed to tackle this challenge, feature extraction techniques such as principal component analysis can be undertaken to condense high-dimensional data into fewer dimensions responsible for maximum variance in the dataset, thereby allowing for better generalizability. Furthermore, feature selection can be undertaken to eliminate noise and incorporate only the features accounting for the variance in output. While these techniques have been used in studies variably to overcome overfitting, external validation has been done only infrequently. A recent systematic review reported that >90% of studies not performing an external validation of their AI/ML algorithms, thereby indicating an urgent need to test for generalizability, albeit in prospectively trials, before its implementation in the clinical practice [78].

Despite the abundance of clinical trials on a yearly basis, inadequate infrastructure, lack of physician training, and incentives are significant hurdles to the process. Moreover, industry-based sponsorship for clinical trials might complicate matters further, as data withholding and the creation of proprietary algorithms might be undertaken for industrial purposes [79]. With a push from the International Committee of Medical Journal Editors (ICMJE) and the National Institute of Health to encourage data sharing and the formulation of repositories such as BigData@Heart, and SWEDEHEART in recent years, the unification of data resources leading to more impactful clinical research might be attainable in the near future [79,80,81].

Furthermore, some AI/ML models suffer from the black box problem, which refers to the limited interpretability of the AI/ML models, with the models becoming increasingly uninterpretable with the increasing complexity of AI/ML algorithms. This becomes a problem in healthcare, as physicians and patients are interested in knowing the result and why the algorithm arrived at a specific result. Not surprisingly, much research has been focused in the past years on making the models more explainable, whereby intelligible approximations of how the model functions (e.g., variable importance plots) are made, helping unzip the black box nature of complex AI/ML models [82].

Apart from the above challenges of model overfitting and lack of interpretability, additional limitations exist specifically to AI/ML models employed for time-series forecasting. Data, once collected, might need to be carefully pre-processed to prevent the model from becoming unstable [83]. Moreover, training AI/ML models for time series forecasting is labor intensive, with neural network-based models requiring substantial computational power. Furthermore, while AI/ML models have been increasingly applied for time series forecasting, there have been few studies directly comparing statistical and AI/ML models, with some showing statistical models to be better than the current AI/ML models [83]. Moreover, besides knowledge of the forecasted variable, researchers are also interested in knowing about the uncertainty surrounding the forecast, which is imperative to making decisions based on forecasts [83]. This becomes especially useful in the clinical setting, whereby creating a parameter of uncertainty (analogous to confidence intervals) can help clinicians decide the significance of a forecasted variable and thereby make insightful medical decisions.

With the demonstrable success of artificial intelligence and the steps taken by the scientific community to address the aforementioned issues, the next decade is set to witness artificial intelligence techniques being employed and tested in randomized controlled trials against standard-of-care modeling, paving the way toward incorporation into clinical practice. Significant strides have already been made with the guidelines being made for the appropriate conduct of trials employing artificial intelligence [84]. Moreover, multiple trials are already underway aimed at testing the role of wearables and AI/ML in improving patient outcomes in HF [53,54,55,56,57,58,59,60]. With AI/ML bound to advance further, future trials could see the unification of structured clinical variables and unstructured medical images to produce robust algorithms, hoping to further our knowledge in understanding disease characteristics and phenogroups within the domain of HF.

## 6. Conclusions

AI/ML and wearable technology, when used together, have the potential to induce a paradigm change in HF management. With few pilot studies showing the effectiveness of AI/ML approaches in conjunction with wearable technology and multiple trials currently underway, the field is only bound to expand. Notable limitations exist at present, which need to be addressed before digitization can be used to its fullest potential. While the scientific community is devising ways at present to address these limitations, the next few years could witness an expansion of the medical armamentarium physicians currently possess, with AI/ML complementing our current tools, henceforth re-enforcing the notion of continuous, physiology-driven monitoring in order to reduce the burden of adverse events, with a focus on hospitalizations, in HF patients.

## Figures and Tables

**Figure 1 diagnostics-12-02964-f001:**
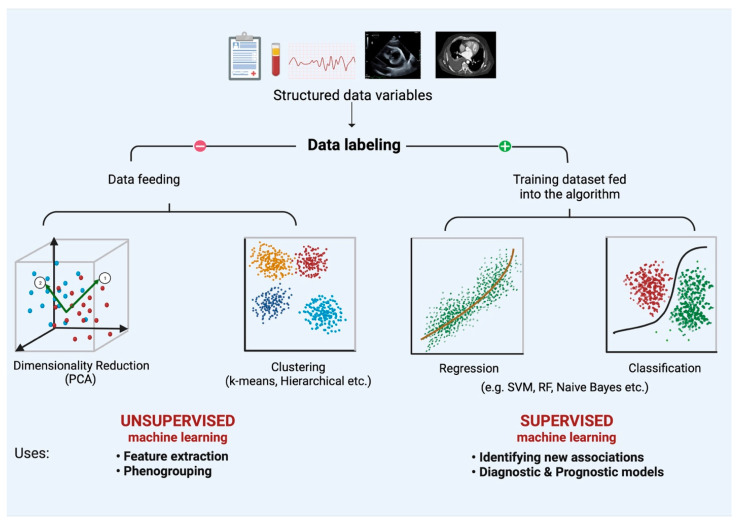
An overview of unsupervised and supervised machine learning. Labeled data can be processed in an unsupervised fashion (i.e., without training data) to understand relationships between variables, clustering, etc. Supervised machine learning, on the other hand, requires a training dataset, which can be used to either infer quantitative relationships (regression models) or classify data into training outcomes (classification). Abbreviations: SVM: Support Vector machines; RF: Random Forest. Adapted from Gautam et al. [13].

**Figure 2 diagnostics-12-02964-f002:**
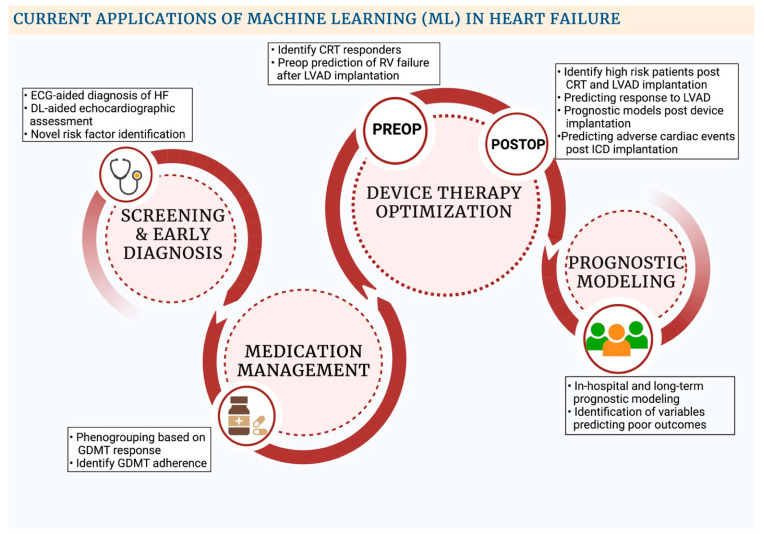
Current applications of machine learning in Heart Failure. Abbreviations: CRT: Cardiac resynchronization therapy; DL: Deep learning; ECG: Electrocardiogram; GDMT: Guideline-directed medical therapy; ICD: Implantable cardiac defibrillator; LVAD: Left Ventricular Assist Device; RV: Right Ventricle.

**Figure 3 diagnostics-12-02964-f003:**
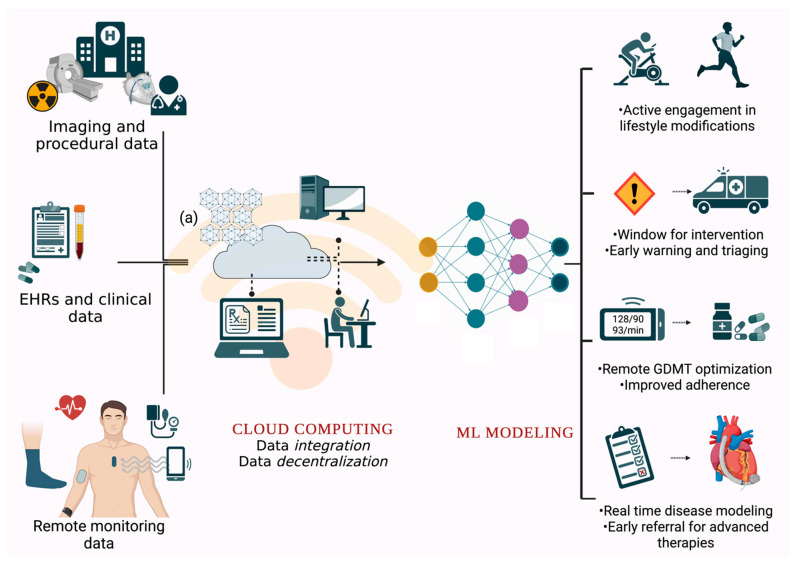
Remote monitoring data, along with data from EHRs and medical images stored on local computers, can be transferred to a cloud computing network, allowing for data integration. Blockchain technology (a) can allow for secure data transfers between systems and allow for data decentralization. Such pools of data, once established, can be used to train powerful ML algorithms, which can be used to guide remote interventions and identify patients needing loser follow-ups. Abbreviations: EHRs: Electronic health records; GDMT: Guidelines directed medical therapy; ML: Machine learning.

**Table 1 diagnostics-12-02964-t001:** AI algorithms used in time-series forecasting.

Algorithms	Description
Support Vector Regression	Supervised ML algorithm based on support vector machines, whereby creation of hyperplane is done in order to minimize the error and maximizes the margin.
k-Nearest Neighbors	Supervised classifier or regression ML algorithm whereby the test variable is classified based on its proximity to k number of data points.
Recurrent Neural network (RNN)	DL algorithm whereby data is passed via multiple layers of neurons (consisting of input, hidden and output layers), with the output from one layer being looped back into the hidden layer, in order to predict sequential data. Associated with the vanishing and exploding gradient problem due to exponential decrease or increase of the gradient during backpropagation, leading to minimal change in adjusted weights in the earlier layers, thereby translating into a short-term memory which can limit its role in larger datasets.
Long short-term memory	Special form of RNN’s (DL algorithms) useful for larger datasets, consisting of a memory unit, comprised of three gates. The forget gate is responsible for screening out irrelevant data, and the output gate is responsible for generation of the new cell state and the hidden state, and the process is repeated over again to yield the final model. Complex model, requiring higher computational power.
Gated recurrent units	Another form of RNN’s (DL algorithms) consisting of two gates-update and the reset gate. The update gate decides the information to be omitted and added and the reset gate is to decide on how much past information can be omitted. Simpler model than LSTM, requiring lesser computational power, while retaining the long-term memory.

Abbreviations: DL: Deep learning; LSTM: Long short-term memory; ML: Machine learning; RNN: Recurrent Neural Network.

**Table 2 diagnostics-12-02964-t002:** Studies evaluating the use of ML models and wearable sensors in HF cohorts.

Author	Study Design and Sample Size (*n*)	ML Model	Results	Limitations
Inan et al. [35] 2018	*n* = 45; single center. SCG signals and ECG signals were analyzed at rest, during 6MWT and 5 min of recovery. GSS was developed using SCG and ECG signals to assess HF state.	GSS developed with the help of K-means clustering	GSS can significantly differentiate between compensated and decompensated HF (*p* < 0.0001). GSS can longitudinally assess improvement in HF status and cardiovascular reserve from admission to discharge (*p* < 0.05).	Differentiation between decompensated and compensated HF groups is subjective and future work is needed to enhance this classification. Investigators were not blinded to the HF state of each patient. Small single center study in a controlled setting.
Shandhi et al. [36] 2022	*n* = 20; measuring changes in PAP and PCWP via SCG signals after vasodilator infusion during RHC.	Globalized (population) regression model developed using logistic regression	Regression model estimated changes in PAP and PCWP in both validation and training sets with good accuracy. SCG signals can be used to track changes in intracardiac pressures non-invasively.	Single center study design with small population size, needs future research to extrapolate results.
Stehlik et al. [39] 2020	*n* = 100; multicenter observational study. Subjects were fitted with a wearable sensor that collected continuous ECG waveform, skin impedance, continuous 3-axis accelerometry, temperature and patient activity/posture data.	Multivariate change index was developed using Cloud-based analytics derived from similarity-based modelling	Multivariate chain index platform was able to detect risk of HF hospitalization with 76% to 87.5% sensitivity and 85% specificity. Clinical alert triggered by personalized machine learning algorithm preceded hospitalization by a median time of 6.5 to 8.5 days. Predictive accuracy to detect impending HF hospitalization was similar to implanted devices.	Non-compliant subjects were excluded from the analysis. Lack of formal testing and validation sets. Observation study mainly on male patients with reduced ejection fraction.
Au-Yeung et al. [44] 2018	*n* = 788; ICD data of patients enrolled in Sudden Cardiac Death-Heart Failure Trial (SCD-HF) was used to automatically predict ventricular arrythmias via heart rate variability (HRV).	RF and SVM	The accuracy of 5-min prediction using RF and SVM was about 0.81 (AUC) whereas 10-s prediction of ventricular arrhythmia was higher with an accuracy of 0.87–0.88.	Real time continuous monitoring requires significant computational resources and would rapidly drain device battery. HRV data employed can be influenced by multiple non-cardiac factors like exercise, anxiety, etc. Rarity of life-threatening ventricular arrythmia increases the difficulty of accurate arrythmia prediction with many false positives.
Joo et al. [45] 2012	*n* = 78 patients; >1000 EKGs from the Spontaneous Ventricular Arrythmia (SVM) database 1.0 from Medtronic ICDs were used to predict VT/VF using HRV analysis.	ANN	ANN models were able to detect VT, VF, VT + VF events with an accuracy of 76.6%, 92.2% and 75.6%, respectively. The normalized areas under the ROC curve of each ANN were 0.75, 0.93 and 0.76, respectively.	Small sample size in the training set was insufficient to ensure statistical classification. Database used had limited pre-VF data, leading to sampling bias. ECGs from single manufacturers were studies; which limits generalizability. ANN require devices with high computational power.
Kim et al. [46] 2022	*n* = 721; A prospective multicenter study aimed at predicting clinically relevant atrial high-rate episodes (AHREs) after pacemaker implantation.	RF, SVMs and extreme gradient boosting	Predictive accuracy of ML models was higher compared to logistic regression-based models (AUC for RF: 0.742, SVM: 0.675 and XgBoost 0.745, vs. logistic regression: 0.669).	Data sets used to develop the validation set were relatively small and contained limited features.
Acharya et al. [38] 2015	*n* = 41 patients; ECG signals from an open access Holter database and normal sinus rhythm database were used to develop a novel integrated index for prediction of SCD.	Decision trees; K-Nearest Neighbor, and SVMs	1. SCD Index had a predictive ability of 92.11%, 98.68%, 93.42% and 92.11% for first, second, third and fourth minutes before the occurrence of SCD, respectively.	1. Small sample size.
Taye et al. [41] 2019	*n* = 55; ECG data from multiple freely available databases was analyzed to predict VF using QRS complex morphology.	ANN, SVM, KNN, RF	Prediction accuracy for VF was significantly higher using QRS complex features compared to HRV features: 98.6% vs. 72% (*p* < 0.05). In addition, sensitivity, and specificity of VF prediction 30 s before occurrence was higher using QRS complex features compared to HRV (AUC 0.99 and 0.71 for QRS complex shape and HRV, respectively).	Small sample size and shorter length of signals before occurrence of VF.
Lee et al. [40] 2019	*n* = 82; early VT prediction ML model was developed using HRV and RRV data from monitors of patients admitted to cardiovascular ICU.	ANN	ML model predicted VT with a sensitivity of 88%, specificity of 82% (AUC: 0.93).	Single center study with small sample size.

Abbreviations: 6MWT: 6-min walk test; ANN: Artificial Neural Network; ECG: Electrocardiogram; GSS: Graph Similarity Score; HF: Heart Failure; HRV: Heart Rate Variability; ICD: Implantable-cardioverter defibrillator; ICU: Intensive Care Unit; KNN: K nearest neighbor; ML: Machine learning; PAP: Pulmonary artery mean pressure; PCWP: Pulmonary capillary wedge pressure; RF: Random forest; RHC: Right heart catheterization; ROC: Receiver Operating Curve; RRV: Respiratory rate variation; SCD: Sudden cardiac death; SCG: Seismocardiogram; SVM: Support Vector Machines; VT: Ventricular tachycardia; VF: Ventricular Fibrillation.

**Table 3 diagnostics-12-02964-t003:** Ongoing clinical trials investigating the application of digital devices and machine learning in HF cohorts.

Clinical Trial	Wearable/Implantable	Description	Current Stage
Activity-Aware Prompting to Improve Medication Adherence in Heart Failure Patients (NCT04152031) [54]	Smartphone	Designing ML-based software algorithms aimed at analyzing daily behavior and to utilize it to improve medication compliance among patients with HF.	Recruitment complete
AIM-POWER study (NCT04191330) [53]	BiovitalsHF	To study the effectiveness of a cloud-based platform (BiovitalsHF) collecting data using remote wearable sensors in improving GDMT adoption among patients with HF.	Recruiting
ASE-INNOVATE study (NCT03713333) [59]	Multiple digital health devices	To study the effectiveness of technology-based visitations (outpatient visit supplemented by focused echocardiography, ECG, and vitals collected by digital devices) on long term cardiovascular outcomes.	Unclear
Heart Failure Monitoring With Eko Electronic Stethoscopes (CardioMEMS) (NCT05080504) [55]	Eko electronic stethoscopes (AI based)	Designing a ML-based algorithm which can correlate Eko stethoscope acoustic and ECG recordings with the pulmonary artery pressure measurements taken via the CardioMEMS device.	Recruiting
Interactive Patient’s Assistant-LUCY (NCT03474315) [56]	Implanted CRT and ICD devices	Designing a ML-based algorithm based on remotely monitored data to determine the parameters in CRT/ICD requiring an ambulatory device clinic visit, overall optimizing long term patient care.	Unclear
LINK-HF2 study (NCT04502563) [57]	Continuous remote monitoring via wearable sensors	To study the impact of remote monitoring on 90-day HF hospitalizations rate in patients with HF.	Recruiting
Validation of Ejection Fraction and Cardiac Output Using Biostrap Wristband (NCT05279066) [58]	Wristband with photoplethysmgraphy sensor	Correlation of ejection fraction and cardiac output measured via AI-powered translation of wristband PPG recordings with echocardiogram and pulmonary artery catheter measurements.	Recruiting
VESTA study (NCT04758429) [60]	Wearable sensor data	Validation of ML algorithm for early detection of HF events via multi-parametric sensor data.	Not yet recruiting

Abbreviations: AI: Artificial Intelligence; ECG: Electrocardiogram; CRT: Cardiac resynchronization therapy; GDMT: Guideline directed medical therapy; HF: Heart Failure; ICD: Implantable cardiac defibrillator; PPG: Photoplethysmography.

## Data Availability

Not applicable.

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
