# Peer review of "Artificial Intelligence, Wearables and Remote Monitoring for Heart Failure: Current and Future Applications"

_diagnostics, 2022, doi:10.3390/diagnostics12122964_

Round 1

Reviewer 1 Report

The study dealt with a current problem comprehensively. Some fixes needed.

1) The title is wrong and incomplete.

2) The study was not only concerned with ML. There is deep learning and machine learning in the work. Therefore, ML should be removed from the title.

3) The study is on wearable technology. However, this is not specified in the title.

4) In Table 1, RNN, LSTM is specified as an ML algorithm. This is not wrong. However, applications such as LSTM and RNN would be more appropriate under the heading of deep learning.

5) Could the following study be a solution to the data shortage for Heart Failure? discuss.

https://doi.org/10.1016/j.bspc.2021.102716

Author Response

We thank the reviewer for the insightful and instructive comments. We would like to address the reviewer’s comments point by point.

The study dealt with a current problem comprehensively. Some fixes needed.

  1. The title is wrong and incomplete.

We would like to thank the reviewer for the comment. As pointed out, we have changed the title to ‘Artificial Intelligence, Wearables, and Remote Monitoring in Heart Failure: Current and Future Applications.’

  1. The study was not only concerned with ML. There is deep learning and machine learning in the work. Therefore, ML should be removed from the title.

We would like to thank the reviewer for the thoughtful comment. As mentioned above, we have removed ML from the title to avoid any confusion.

  1. The study is on wearable technology. However, this is not specified in the title.

We would like to thank the reviewer for the thoughtful comment. As mentioned in point 1, we have changed the title to  ‘Artificial Intelligence, Wearables, and Remote Monitoring in Heart Failure: Current and Future Applications’ in order to give the reader a complete idea about the manuscript details.

  1. In Table 1, RNN, LSTM is specified as an ML algorithm. This is not wrong. However, applications such as LSTM and RNN would be more appropriate under the heading of deep learning.

We would like to thank the reviewer for the thoughtful comment. As pointed out, we have changed the title of Table 1 to ‘AI algorithms used in time-series forecasting’. Moreover, under the description for Recurrent Neural Networks (RNN), Long Short Term Memory (LSTM), and Gated Recurrent Units (GRUs), we have mentioned them to be a type of deep learning algorithm for a better understanding.

  1. Could the following study be a solution to the data shortage for Heart Failure? discuss.

https://doi.org/10.1016/j.bspc.2021.102716

We would like to thank the reviewer for the excellent suggestion. Apart from the conversion of numeric data to image form, which can then be processed by Convolutional neural networks (CNNs), Natural language processing (NLPs) can also help convert data into a form, which can then be fed into DL algorithms. To deliver the same, we have added the following lines in the manuscript:

‘Moreover, while tons of data are stored in electronic health records, such troves of data are not readily employable and often need to be converted and restricted before being fed into an ML algorithm. More recently, Aslan et al. devised a convolutional neural network (CNN) based novel approach to convert the numerical data into images, which can then be accurately classified for the purpose of diagnosing HF. Such an approach, along with data extraction methods that use natural language processing algorithms, can help expand the current data pool and, thereby, lead to improved performance of the current diagnostic and prognostic models.’

Thank you very much for your consideration of this manuscript for publication. Again, we express our sincere gratitude for your time and valuable feedback.

Yours Sincerely,

Subhi J. Al’Aref, MD, FACC

(Corresponding author)

Reviewer 2 Report

This is an interesting narrative review on the beneficial effects of artificial intelligence on the management of heart failure but with the following limitations:

It does not explain whether the application of these interventions is cost/effective. This aspect should be further explored as it is very relevant from a practical point of view.

What about patients with cognitive impairment, severe visual/hearing impairment or poor social support, and would they also benefit from artificial intelligence?

Although this is a narrative review, it would be very useful if the authors could explain their literature search strategy and justify why they have added these articles and not others in their review.

Once these aspects have been resolved, it can be published.

Author Response

We thank the reviewer for their insightful and instructive comments. We would like to address the reviewer's comments point by point

This is an interesting narrative review on the beneficial effects of artificial intelligence on the management of heart failure but with the following limitations:

  1. It does not explain whether the application of these interventions is cost/effective. This aspect should be further explored as it is very relevant from a practical point of view.

We want to thank the reviewer for the excellent suggestion. Indeed it's very important from a practical aspect for the wearables to be cost-effective. While a recent study published showed the cost-effectiveness of photoplethysmography to detect atrial fibrillation in individuals, there is a paucity of studies testing the cost-effectiveness of devices when coupled to ML in the realm of HF. While we have addressed that as mentioned below, we have also highlighted the financial challenges associated with wearables and the potential solutions in section 5.3:

‘ There is a potential for a 'digital divide' with the dawn of wearable sensors in HF. Apart from the clinical heterogeneity observed in patients with HF, socioeconomic status (SES) plays a paramount role, with studies showing more than a 50% increase in HF risk with lower SES. While wearable technology has the potential to transform cardiovascular care, it can increase health inequity, owing to the need for adequate digital affordability and literacy associated with wearable technology. For instance, in a survey of 4551 adults in the United States, individuals with a higher annual household income (defined as $75,000 or more per year) were more likely to use wearables than individuals with a household income of less than $20000 per year (OR 2.6; 95% CI [1.39-4.86], p<0.001). Incorporating the device-related costs into the healthcare insurance benefit packages is crucial for narrowing the digital divide based on economic disparities. Moreover, loaner digital wearables can help mitigate the financial implications on healthcare, provided checks are in place to ensure the return of digital devices. The novel iShare program utilizing loaner iPhones and smartwatches in 200 patients demonstrated augmented participation of patients without access to smartphones in remote monitoring following an acute myocardial infarction. However, with a modest return percentage of 72% at the end of the 30-day follow-up period, a system of checks and incentives is needed for its widespread implementation. Nevertheless, the cost-effectiveness of wearable devices coupled with AI modalities needs to be studied in HF, with such devices being previously proven to be cost-effective in the screening of other cardiac pathologies, such as atrial fibrillation. With pulmonary arterial pressure monitoring being shown to be cost-effective in multiple studies, complementing remote monitoring with AI is likely to improve the cost-benefit ratio, although it yet needs to be proven across a spectrum of population cohorts.

  1. What about patients with cognitive impairment, severe visual/hearing impairment, or poor social support, and would they also benefit from artificial intelligence?

We want to thank the reviewer for the excellent suggestion. It is challenging at present for certain population strata to adopt digital technology, with age and socioeconomic status being the key factors playing a role in that. In order to address the above-mentioned point, we have added the following paragraph in section 5.3:

‘A lack of digital literacy can not only impede the adoption of digital technology but can also impact patient compliance with data transmission. Not surprisingly, individuals possessing college degrees or greater have been shown to be more receptive to the adoption of digital technology when compared to people with a lesser educational background. Moreover, elderly individuals are less likely to adopt the usage of digital technology and also to comply with regular data transmission. While it is essential to devise ways to improve the usage of wearables in this at-risk population, how these measures can be implemented in patients with neurocognitive disorders and visual/hearing impairment remains challenging. ‘

  1. Although this is a narrative review, it would be very useful if the authors could explain their literature search strategy and justify why they have added these articles and not others in their review.

We want to thank the reviewer for the suggestion. While we have tried to be as thorough as possible in our literature review strategy (utilizing PubMed MeSH keyword search strategy and Google scholar), the article was, by no means, a systematic review of studies employing AI in remote monitoring in heart failure. While it would certainly be exciting to perform a systematic review on a specific aspect of remote monitoring in HF, our goal, via this narrative review, was to provide an overview of the current applications of artificial intelligence and the future aspects in the field of remote monitoring in HF.

Thank you very much for your consideration of this manuscript for publication. Again, we express our sincere gratitude for your time and valuable feedback.

Yours Sincerely,

Subhi J. Al’Aref, MD, FACC

(Corresponding author)

Round 2

Reviewer 1 Report

The authors have improved the article, it is acceptable.